# Two-Level Actor-Critic Using Multiple Teachers

**Su Zhang**                                                          *su.zhang2@wsu.edu*
*Washington State University*
*Pullman, United States*

**Srijita Das**                                                       *srijita1@ualberta.ca*
*University of Alberta*
*Edmonton, Canada*

**Sriram Ganapathi Subramanian**                    *sriram.subramanian@vectorinstitute.ai*
*Vector Institute*
*Toronto, Canada*

**Matthew E. Taylor**                                          *matthew.e.taylor@ualberta.ca*
*University of Alberta*
*Alberta Machine Intelligence Institute*
*Edmonton, Canada*

**Reviewed on OpenReview:** *https://openreview.net/forum?id=LfQ6uAVAEo*

## Abstract

Deep reinforcement learning has successfully allowed agents to learn complex behaviors for many tasks. However, a key limitation of current learning approaches is the sample-inefficiency problem, which limits performance of the learning agent. This paper considers how agents can benefit from improved learning via teachers' advice. In particular, we consider the setting with multiple sub-optimal teachers, as opposed to having a single near-optimal teacher. We propose a flexible two-level actor-critic algorithm where the high-level network learns to choose the best teacher in the current situation while the low-level network learns the control policy.

## 1 Introduction

Reinforcement learning (RL) has been successful in a variety of domains ranging from solving difficult games like Go (Silver et al., 2016) and Dota (Berner et al., 2019) to more important problems like drug discovery (Gottipati et al., 2021). Most of these domains are characterized by high dimensional state and continuous action space. However, sample inefficiency is one of the major challenges of applying these algorithms to real-world tasks like robotics and healthcare (Ibarz et al., 2021). Millions of physical interactions with the environment are typically infeasible due to hardware degradation, safety issues, etc.

To address improved sample efficiency, rather than forcing agents to learn from scratch, domain knowledge from humans or existing agents can be leveraged in various ways (Da Silva & Costa, 2019). Reward shaping (Ng et al., 1999) and policy shaping (Griffith et al., 2013) are two popular techniques that can bias the learning agent's reward or policy, respectively, for faster learning. A different method is action advising (Torrey & Taylor, 2013), where a human or agent *teacher* can tell a learning *student* what action to take in a state. This approach is particularly useful because the teacher and the student agent can have completely different internal representations (which is not generally true in transfer learning settings (Taylor & Stone, 2009)).

Many existing approaches can leverage advice in RL from a single, near-optimal teacher (Hester et al., 2018; Warnell et al., 2018; Bignold et al., 2021). In this work, we consider settings where a student can receive

*action advice from multiple teachers.* This setting can be particularly appropriate when different teachers have different skills (e.g., teachers may perform well in different parts of the state space or perform different sub-tasks). In addition, we also consider that teachers may be suboptimal or even random. This allows us to leverage teachers that perform well while not being hurt (much) by teachers that perform poorly. This paper focuses on the question, "When should the student listen to which teacher?" to effectively use the best teacher's policy for a given state or to decide not to listen to any teacher.[1]

As a motivating example, let us consider the example of an agent learning how to drive. In this scenario, one teacher (T1) could know how to effectively do basic driving skills while another teacher (T2) knows how to do night-driving specifically. These teachers could then collectively assist the student agent in the appropriate parts of state-space to achieve the sub-tasks, despite not having knowledge about the entire driving task. Our novel approach would allow a student agent to learn to query T1 more frequently for advice regarding basic driving skills and T2 for night-driving protocols while learning to drive safely. While the above example is similar to hierarchical RL, in this work, the main task need not be divided into sub-tasks and the teachers could be sub-optimal.

**Contributions:** Our novel approach:

1. learns from the action advice of multiple teachers by extending the actor-critic framework (Mnih et al., 2016) to a two-level framework where the high level learns how to pick teachers to listen to, and the low level learns the agent's policy
2. is a general framework that can leverage commonly available sub-optimal or complimentary teachers
3. is robust to the quality of teachers, learning even in the presence of poor teachers
4. succeeds empirically, relative to existing methods in a variety of tasks like Hopper and Pick & Place

Our work is inspired by Two-Level Q-Learning (TLQL) (Li et al., 2019b) but is different in two ways: (1) it can be applied to a variety of tasks having both continuous and discrete action-space and (2) is more resistant to teachers of different qualities ranging from fully optimal to partially sub-optimal teachers. In addition, by using two actor-critic networks, our approach can be easily incorporated into any existing actor-critic algorithm.

## 2 Background and Prior Work

### 2.1 Background

**Reinforcement Learning:** RL (Sutton & Barto, 2018) tasks are often formalized as Markov decision processes (MDPs). The MDP is modelled as an infinite horizon system $(\mathcal{S}, \mathcal{A}, r, T, \gamma)$ where $\mathcal{S}$ is the state space, $\mathcal{A}$ is the action space, $r$ is the immediate reward, $T$ is the state-transition probability, and $\gamma \in [0, 1)$ is the discount factor. An agent receives a state $s_t \in \mathcal{S}$ from the environment at time-step $t$, takes action $a_t \in \mathcal{A}$, and after interacting with the environment, gets a reward $r_{t+1}$ and transitions to the next state $s_{t+1} \in \mathcal{S}$. The agent maximizes the expected return $\mathcal{R}_t = \sum_{k=0}^{\infty} \gamma^k r_{t+k}$, the discounted sum of rewards from time-step $t$, and acts according to $\pi$.

**Advantage actor-critic:** The advantage actor-critic (A2C) method is the synchronous variant of the asynchronous advantage actor-critic (A3C) (Mnih et al., 2016) algorithm. A2C consists of two networks: the policy network $\pi(a_t|s_t; \theta)$ and the value network: $V(s_t; \theta')$. The policy and the value network can either have shared parameters or can be entirely different networks with parameters $\theta$ and $\theta'$, respectively. The loss function optimized by the policy network is $\mathcal{L}(\theta') = \log \pi(a_t|s_t; \theta)(\mathcal{R} - V(s_t; \theta'))$, where $\mathcal{R}$ is the n-step return calculated as $\mathcal{R} = \sum_{i=1}^{n-1} \gamma^i r_{t+i} + \gamma^n V(s_{t+n}; \theta')$. The state value function $V(s_t; \theta')$ is called baseline (Williams, 1992) and is used to reduce the variance of the policy estimate. The value network minimizes the squared loss $(\mathcal{R} - V(s_t; \theta'))^2$ to update the parameters.

---

[1]This work assumes teachers do not learn and are not antagonistic, which are instead left to future work.

## 2.2   Related Work

Early approaches to incorporating multiple teachers include aggregating multiple teachers into a single teacher by using weighted mixture distributions (Jacobs et al., 1991) or aggregating the action distributions for each teacher (Hinton, 2002). Recent imitation learning work includes combining multiple suboptimal teachers using online learning (Cheng et al., 2020), bi-level optimization to learn the confidence of experts from demonstrations (Zhang et al., 2021), and using advantage functions to decide and take advice from the best teacher in a state (Li et al., 2019a). Our proposed work is a reinforcement learning framework that incorporates action-advice from multiple teachers and differs from imitation learning.

*Knowledge reuse from multiple teachers (single-agent):*  Different measures like similarity and regret bounds (Fernández & Veloso, 2006; Azar et al., 2013; Zhan et al., 2016) have been used in prior work to incorporate advice or policies from multiple teachers into single-agent learning. However, the effectiveness of these approaches on complex and continuous control tasks has not yet been demonstrated. Gimelfarb et al. (2018) use a Bayesian model to combine potential based reward/value functions from multiple teachers. Our approach uses action-advice, which is more practical than leveraging value functions from multiple experts. Seita et al. (2019) proposed to select the best suited teacher which is at least on par or slightly better than the student agent. However, their work could not accommodate partial teachers, and they query teachers to seek demonstrations. Most similar to our problem setting is AC-Teach (Kurenkov et al., 2019), which uses an ensemble of suboptimal teachers that could be partial or contradictory in a Bayesian actor-critic framework to guide the student agent. AC-Teach uses strategies to be able to listen to the same teacher for a longer period, which is unsuitable for tasks where different teachers might give useful advice in different parts of the task.

*Knowledge reuse from multiple teachers (multi-agent):*  A lot of work has been done to reuse knowledge from other agents in a multi-agent setting (Da Silva & Costa, 2019). Da Silva et al. (2017) propose a framework in which the agents learn when to seek or give advice based on heuristics like state-visitation frequency. Omidshafiei et al. (2019) provide a more general framework where the teacher agent decides when and what to advise, and the student agent learns to utilize the received action-advice. Kim et al. (2011) scale prior work to larger and more complex domains by improving the teacher's policy by considering the effect of advice on the student's learning. All the above-mentioned work shares similarities because the role of the teacher and student is not fixed and varies depending on their confidence or uncertainty about a task. Our work focuses on single-agent learning with multiple sub-optimal fixed teachers.

*Layered learning:*  Li et al. (2019c) map the problem of selecting multiple source policies as an option-learning framework to accelerate contextual policy reuse in transfer learning. Zhang & Whiteson (2019) uses a two-level actor-critic framework for option learning. However, their framework doesn't accommodate advice from teachers. Our method is *not an option learning framework* and both the networks use the same underlying MDP in contrast to the former which uses two different augmented MDP at each layer. Our work is also inspired by Li et al. (2019b), who use a two-level Q-learning algorithm to select the best teacher and its action advice, respectively. However, their evaluations were limited to tabular Q-learning and smaller discrete domains. They also propose a variant with the high-level network modeled as DQN, which still could not be applied to continuous control tasks. Another relevant framework (Xie et al., 2018) uses an external controller as a teacher to guide the deep RL agent and uses DQN to select the relevant policy to be used. However, this was limited to policy selection between a single teacher and student agent. *The effect of the number of teachers and the quality of advice from multiple teachers, including noisy teachers, was not studied systematically in any prior work.* A recent work (Subramanian et al., 2022) uses a similar two-level actor-critic framework with sub-optimal teachers in fixed roles. However, the setup is multi-agent single teacher as opposed to our single-agent multi-teacher setup. Our proposed work can apply to both discrete and continuous control tasks and scales well with a number of noisy teachers and advice quality.

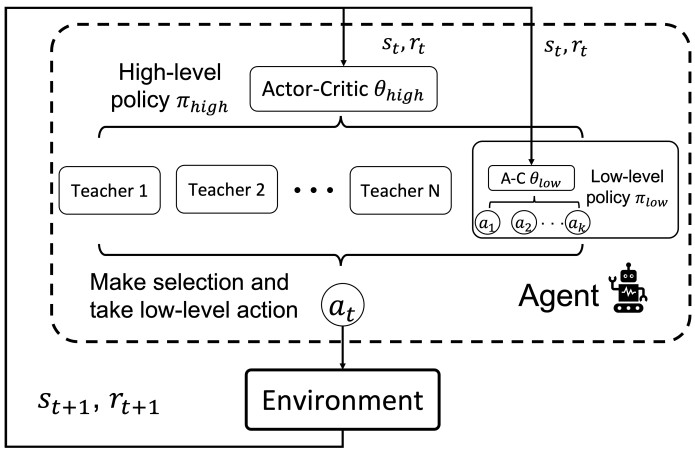

Figure 1: Two-Level Actor-Critic Structure

## 3 Two-Level Actor-Critic Using Multiple Teachers

We introduce a Two-Level Actor-Critic (TL-AC) method to learn from multiple teachers. This method extends the actor-critic algorithm to a two-level network structure, with a single critic for both levels, enabling the agent to leverage teachers with different expertise and advice quality.

### 3.1 Problem Statement

**Given:** A set of (sub)optimal teachers as denoted by $E_0 = \{e_1, e_2, \cdots, e_N\}$ where $|E_0| = N$, with a set of corresponding policies denoted by $\pi_E = \{\pi_{e_1}, \pi_{e_2}, \cdots, \pi_{e_N}\}$.

**Objective:** Train the learning agent by leveraging action-advice from multiple teacher policy set $\pi_E$ when useful to make the agent learn a good policy with fewer environment interactions.

**Assumptions:** We consider a single-agent learning problem with multiple teachers. The teachers are fixed during the agent training process. The teachers can be either human teachers or agents. The teachers can always provide action advice to the student about the current state, and the student need not know the decision mechanics behind the advice. The action advice could be optimal or sub-optimal, depending on the teachers' expertise. We explicitly assume in this work that communication between the student and teacher is unlimited, free, and uncorrupted. Hence, teachers are always available in this setup.

The multiple teachers used in our approach can be pre-trained agents, classifiers trained from demonstrations, human teachers, etc. At each time-step $t$, each individual teacher $e_i$ provides action advice based on the state vector $s_t$. The student agent will then choose an action to perform according to a certain policy, which can be either its own policy or the policy of any one of the several teachers in the set $E$. The goal of the learning agent is to maximize the reward. Meanwhile, the optimal policy of how to wisely listen to multiple teachers should also be found.

### 3.2 Algorithm and Methodology

Figure 1 shows how the two-level actor-critic structure, detailed in Algorithm 1. The algorithm collect a minibatch $\mathbf{M}$ with Function $collect(\pi_{high}, \pi_{low}, E, m)$ (Lines $11 - 23$). The high-level policy is a mapping from states to teachers, $\pi_{high}(s) : S \rightarrow E$, where $E = E_0 \cup Agent$. The high-level policy also considers the agent itself. The low-level policy is a mapping from states to actions, $\pi_{low}(s) : S \rightarrow \mathcal{A}$. During collecting the transactions, at state $s_t$, the behavior policy first follows the high-level policy $\pi_{high}$ to choose which teacher to listen to (Line 14). Once a teacher is selected, denoted $e_t$, the algorithm selects an action according to this teacher's policy: $a_t \leftarrow \pi_{e_t}$ (Line 18). If the agent is selected by the high-level policy, the algorithm selects an action with the agent's own policy $a_t \leftarrow \pi_{low}$ (Line 16). The resulting transition, $\langle s_t, a_t, s_{t+1}, r_t, e_t \rangle$, is saved

to the minibatch (Line 19). To update those two networks, in each epoch, the advantage is calculated with Eq. 2 (Line 7). This advantage calculates the low-level loss with Eq. 3 and updates the low-level network (Line 7), calculating the high-level loss with Eq. 7 and updating the high-level network (Line 8).

---

**Algorithm 1: $\underline{\text{T}}$wo-$\underline{\text{L}}$evel $\underline{\text{A}}$ctor-$\underline{\text{C}}$ritic using Multiple Teachers**

**Input:** Teachers set $E_0$, union with the RL agent $E = E_0 \cup Agent$, minibatch size $m$, epoch # $N$

1 Initialize: low-level policy $\pi_{low}$, high-level policy $\pi_{high}$

2 **for** *each episode* **do**

3    **repeat**

4       $\mathbf{M} \leftarrow$ collect($\pi_{high}$, $\pi_{low}$, $E$, $m$)               $\triangleright$ Collect rollouts

5       **for** *epochs in $N$* **do**

6          $\mathbf{M} = \{\tau_i = (s_t, a_t, r_t, s_{t+1}, e_t)\}_{i=1}^m$       $\triangleright$ Update with minibatch $\mathbf{M}$

7          Calculate advantage $A(s_t, a_t) = r_t + \gamma V_{\pi_{low}}(s_{t+1}) - V_{\pi_{low}}(s_t)$

8          Update low-level network $\theta_{low}$ by optimizing the loss$^2$ $\mathcal{L}(\theta_{low}) = \log \pi_{low}(a|s; \theta_{low}) A(s, a)$

9          Update high-level network $\theta_{high}$ by optimizing the loss$^2$ $\mathcal{L}(\theta_{high}) = \log \pi_{high}(e|s; \theta_{high}) A(s, a)$

10    **until** *episode end*;

11 **Function** collect($\pi_{high}$, $\pi_{low}$, $E$, $m$):

12    Get state $s_t \sim$ Env

13    **repeat**

14       $e_t \leftarrow \pi_{high}(e|s_t)$                      $\triangleright$ Select teacher $e_t$ with high-level policy $\pi_{high}$

15       **if** $e_t$ *is Agent* **then**

16          $a_t \leftarrow \pi_{low}(a|s_t)$               $\triangleright$ Select action with low-level policy $\pi_{low}$

17       **else**

18          $a_t \leftarrow \pi_{e_t}(a|s_t)$               $\triangleright$ Select action with teacher policy $\pi_{e_t}$

19       Execute action $a_t$ and get transaction $\tau_{t+1} = (s_t, a_t, s_{t+1}, r_t, e_t)$

20       Save transaction $\tau_{t+1}$ to minibatch $\mathbf{M} = \mathbf{M} \cup \{\tau_{t+1}\}$

21       $s_t \leftarrow s_{t+1}$

22    **until** $\mathbf{M}$ *reach size $m$*;

23    **return** $\mathbf{M}$

---

**Low-level Policy: Select Action** The low-level policy network is the first actor-critic network, parameterized with $\theta_{low}$. The low-level policy $\pi_{low}(a|s; \theta_{low})$ maps the states to a probability distribution over actions. The discounted value at state $s$ under the low-level policy $\pi_{low}$ is

$$V_{\pi_{low}}(s) := \mathbb{E}\left[\sum_{t=0}^{\infty} \gamma^t r(s_t, a_t)|s_t = s\right] \tag{1}$$

The advantage of taking action $a_t$ at state $s_t$ is

$$A(s_t, a_t) := r(s_t, a_t) + \gamma V_{\pi_{low}}(s_{t+1}) - V_{\pi_{low}}(s_t) \tag{2}$$

The objective is to maximize the agent's value over all states and find the low-level optimal policy with the loss function $^2$

$$\mathcal{L}(\theta_{low}) = \log \pi_{low}(a|s; \theta_{low}) A(s, a) \tag{3}$$

**High-level Policy: Select Teacher and Take Advice** The high-level policy network is the second actor-critic network, parameterized with $\theta_{high}$. The high-level policy $\pi_{high}(e|s; \theta_{high})$ maps the states to a probability distribution over teachers. The high-level network estimates the expected return of selecting each teacher $e \in E$. The reward for a teacher is the same as the environment reward that the agent received by executing this teacher's action advice. Hence, we have

$$r(s_t, e_t) = r(s_t, a_t|e_t) = r(s_t, a_t) \tag{4}$$

---

$^2$The subscript '$t$' is dropped from notation for convenience because the loss-function is defined with respect to a single mini-batch.

The discounted value at state $s$ under the high-level policy $\pi_{high}$ is

$$
\begin{aligned}
V_{\pi_{high}}(s) &:= \mathbb{E}\left[\sum_{t=0}^{\infty} \gamma^t r(s_t, e_t)|s_t = s\right] \\
&= \mathbb{E}\left[\sum_{t=0}^{\infty} \gamma^t r(s_t, a_t)|s_t = s\right] \\
&= V_{\pi_{low}}(s).
\end{aligned}
\tag{5}
$$

The advantage of choosing teacher $e_t$ at state $s_t$ at the high-level could be written as

$$
\begin{aligned}
A(s_t, e_t) &:= r(s_t, e_t) + \gamma V_{\pi_{high}}(s_{t+1}) - V_{\pi_{high}}(s_t) \\
&= r(s_t, a_t) + \gamma V_{\pi_{low}}(s_{t+1}) - V_{\pi_{low}}(s_t) \\
&= A(s_t, a_t)
\end{aligned}
\tag{6}
$$

This indicates that the two level networks are using the same critic for estimating the value function of the states. The objective is to maximize the value over all states and find the optimal high-level policy with the loss function [2]

$$
\mathcal{L}(\theta_{high}) = \log \pi_{high}(e|s; \theta_{high}) A(s, a)
\tag{7}
$$

In this way, we use one single critic for both high-level and low-level policies instead of separate ones. This single critic estimates one value function based on the current state. A similar reduction was shown by Zhang & Whiteson (2019) in an option-learning framework.

## 4 Experiments

The experiments are designed to investigate the following two research questions:

**R1:** Can the two-level actor-critic method handle a mixture of sub-optimal teachers relative to existing methods?

**R2:** Can the two-level actor-critic method incorporate multiple partial teachers with different areas of expertise?

### 4.1 Experimental Settings:

We present experimental results on a discrete control task (DoorKey) and two continuous control robotics tasks (Hopper and Pick & Place) to demonstrate the effectiveness of our method. The environment details for our experiments are as below:

**1. Door & Key Environment** is a grid room environment from the Minimalistic Gridworld Environment (MiniGrid) (Chevalier-Boisvert et al., 2018). The rooms are separated by a wall: the agent needs to use the key to open the door and enter the other half to get to the target grid. The state is a 3-tuple vector, and there are 6 discrete actions. A (sparse) positive reward is given when the goal is reached; otherwise, it is 0.

**2. Hopper** is a two-dimensional one-legged robot, and the task is to hop forward as far as possible. We use the HopperPyBulletEnv-v0 from the PyBullet Gymperium (Ellenberger, 2018–2019) for the experiments. The 15-dimensional state space and the 3-dimensional action space are both continuous. The dimension of state space is 15, and the dimension of action space is 3. The dense reward is calculated based on its current potential, alive bonus, and power cost.

**3. Pick & Place** is a robotic manipulation task from the Fetch environments (Plappert et al., 2018). The goal is to pick up a box and place it at a (fixed) target location. The observation space is a (continuous) 25-dimensional vector, and the action space is a 4-dimensional continuous vector to control the end effector and the gripper. The dense reward is based on the distance between the box and the target.

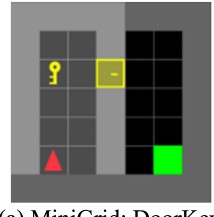
(a) MiniGrid: DoorKey

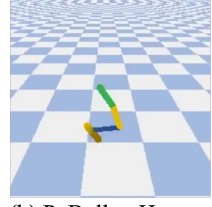
(b) PyBullet: Hopper

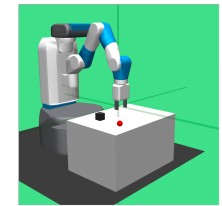
(c) MuJoCo: Pick & Place

Figure 2: Experimental Domains

**Baselines:** We use the A2C algorithm without any advice as the baseline. We also benchmark against the DQN variant Two-level Q-learning (DQN-TLQL) (Li et al., 2019b) and AC-Teach (Kurenkov et al., 2019), both of which can learn from multiple teachers. DQN-TLQL does not support continuous control. Hence, experiments could not be performed on Hopper and Pick & Place domains for this benchmark. We only consider state-of-the-art algorithms with advice from multiple teachers for fair comparison.

**Implementation Details:** The A2C baseline uses the default setting of the Stable Baseline3 (Raffin et al., 2019). For DQN-TLQL, since we do not have access to the original codebase, we refer to the Stable Baseline3 and implement the DQN version of the Two-level Q-learning algorithm as described in (Li et al., 2019b). The DQN-TLQL and A2C hyperparameters are listed in Appendix B. The implementation and parameter settings of AC-Teach are from the original codebase of Kurenkov et al.'s work (Kurenkov et al., 2019)[3].

**Teacher Set Details:** To construct the teacher set[4], we first obtain a near-optimal policy, then introduce different levels of noise[5] to this policy to get *good teacher*, *noisy teacher*, and *random teacher*. We also construct *partial teachers* for DoorKey and Pick & Place, who could only provide good advice on partial state space. Details of the teacher set for each experiment are in Appendix A.1.1, A.2.1, and A.3.1.

**Performance Metric:** We consider two widely-used evaluation metrics in RL, which have been proven to be robust in evaluating the performance of learning from teachers approaches: 1) **Jumpstart**, the improvements of initial performance over the benchmark agent (Taylor & Stone, 2009) and 2) **Mean training reward**, the plotted learning curve as well as the standard deviation (Sutton & Barto, 2018).

To guarantee that all algorithms are compared with the same number of updates, the batch size and gradient steps setting are adjusted accordingly for both DQN-TLQL and TL-AC. We report the training performance of all the algorithms for fair comparison.[6] All DQN-TLQL, A2C, and TL-AC results are averaged over 5 seeds. Due to the limited computing resources and the observation of its relatively steady performance, the AC-Teach results are averaged over 3 seeds.

### 4.2 Results and Analysis

This section presents results from the three domains. We address R1 for all three domains. For R2, since it's difficult to obtain partial teachers with pre-trained agents in Hopper, we only conduct experiments with the DoorKey and the Pick & Place domain.[7]

**R1: Can TL-AC handle a mixture of sub-optimal teachers relative to other existing methods?** To answer **R1**, we provide advice from four sets of three teachers of varying quality to the RL learning agents, including the baseline agents.[8] Increasing the number of good teachers will not provide extra help in accelerating learning. The results of DoorKey are shown in Figure 3: the A2C baseline does not learn in this domain without any external bias because of sparse reward; hence, intrinsic motivation related approaches

---

[3]https://github.com/StanfordVL/ac-teach

[4]Teachers could be humans or agents. In this work, we use a fixed set of trained agents.

[5]We currently experimented with the fixed percentage of random noise and Gaussian action noise of a certain scale.

[6]In (Kurenkov et al., 2019), the authors report the testing performance of AC-Teach without the teachers.

[7]Because the state of Hopper only measures the progress of each step (velocity), not the absolute coordinate, it is difficult to find an explicit way to split an optimal teacher into two partial teachers.

[8]The underlying assumption is that a single good teacher is sufficient to solve this task. However, it might be difficult to identify them from a pool of teachers.

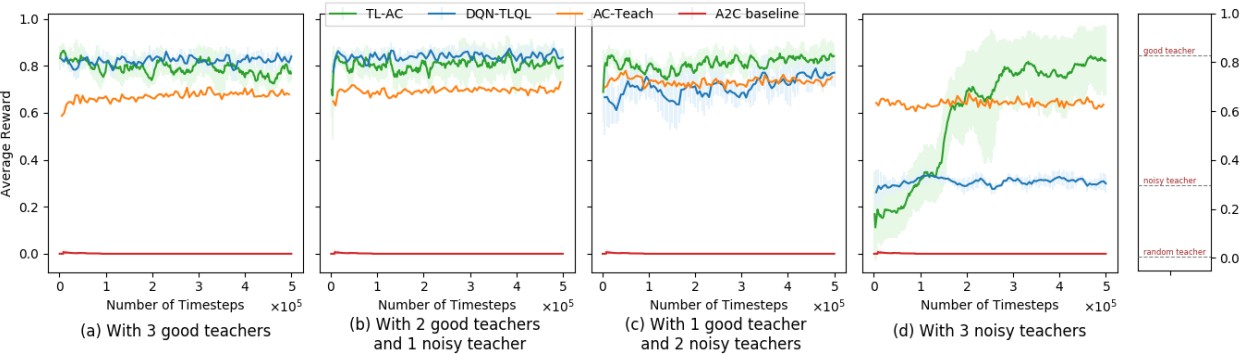

(a) With 3 good teachers
(b) With 2 good teachers and 1 noisy teacher
(c) With 1 good teacher and 2 noisy teachers
(d) With 3 noisy teachers

Figure 3: DoorKey results show the performance impacts of different teachers. All three algorithms perform well with 3 good teachers. With 3 noisy teachers, our TL-AC method (green line) starts with lower initial performance but gradually catches up and converges to better performance.

are used for this domain (Chevalier-Boisvert et al., 2018). With 2 and 3 good teachers, TL-AC has a similar performance level to DQN-TLQL and performs better than AC-Teach (Figures 3(a) and (b)). When there are two noisy and one good teacher (Figure 3(c)), TL-AC could effectively outperform both DQN-TLQL and AC-Teach. When only noisy teachers are provided (Figure 3(d)), compared to the settings in Figure 3(a)(b)(c), the performance of AC-Teach drops slightly but has a higher jumpstart, TL-AC starts with lower initial performance but gradually converges to near-optimal performance. DQN-TLQL does not learn from a combination of only noisy teachers. For TL-AC, in the beginning, the agent needs to either listen to noisy teachers' advice or act with its own policy, which is random at this stage. When the student's own policy (i.e., the low-level policy) gradually improves as training progresses, it will tend to choose its own policy over the noisy teachers'. Figure 3(d) shows more pronounced learning progress as compared to Figures 3 (a)(b)(c). AC-Teach follows one single selected (teacher) policy for a longer time horizon than TL-AC and switches the selection with an estimated confidence-based scheme (Kurenkov et al., 2019). This may explain why its training performance looks more steady but shows fewer improvements over time. This observation holds for all the experiments in this section.

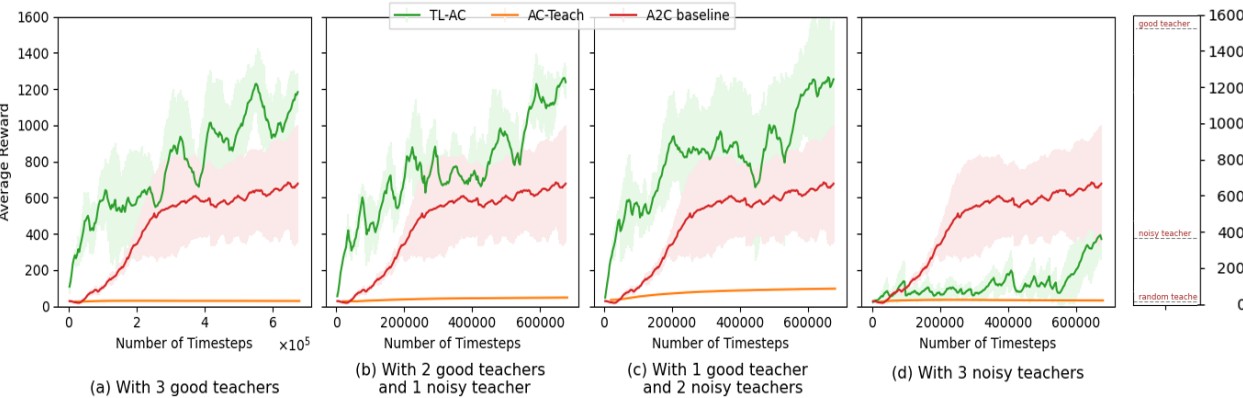

(a) With 3 good teachers
(b) With 2 good teachers and 1 noisy teacher
(c) With 1 good teacher and 2 noisy teachers
(d) With 3 noisy teachers

Figure 4: Hopper results show the performance impacts of providing multiple teachers of different qualities. For (a)(b)(c) scenarios, TL-AC could always surpass the A2C baseline. For (d), TL-AC is affected by the noisy teachers but gradually learns and improves its policy. Comparatively, AC-Teach performs poorly in this task.

We also answer **R1** in Hopper (Figure 4) and Pick & Place (Figure 5). In scenarios that provide at least one good teacher, TL-AC could always learn from it and achieve higher performance. However, when there are only noisy teachers, the performance of TL-AC decreases. In the Hopper task, TL-AC could gradually learn

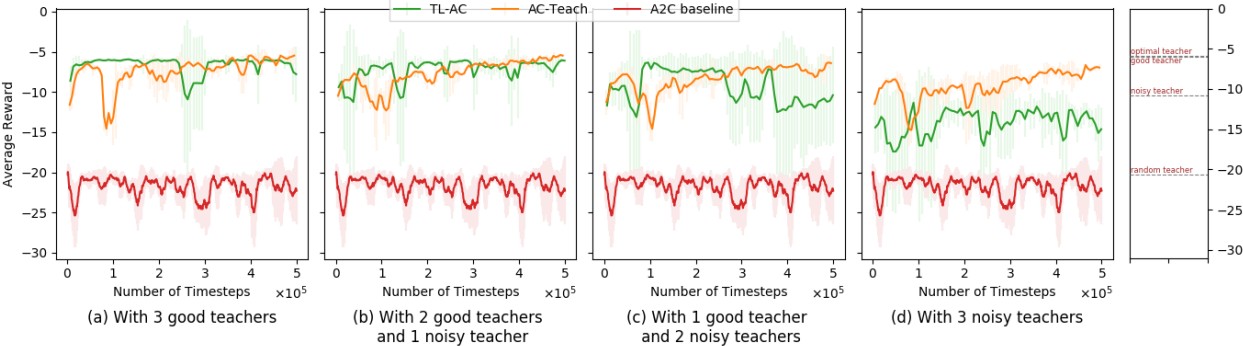

Figure 5: Pick & Place results show the performance impacts of providing multiple teachers of different qualities. For (a)(b)(c), TL-AC has a similar performance with AC-Teach, while it could be affected by the noisy teachers in (d).

and improve its policy over time[9], while in the Pick & Place task, it is harder to recover. A possible reason for this performance drop could be that, during the initial learning phase, the noisy teachers' advice could mislead the exploration and the critic (value estimation) update, hence interfering with both the high-level and low-level policy of TL-AC. AC-Teach performs better than TL-AC in Pick & Place with 3 noisy teachers but consistently underperforms A2C and TL-AC in Hopper.

To summarize, the DoorKey experiment showed that TL-AC can effectively learn from multiple teachers of different quality as compared to other baselines in discrete tasks and is especially useful when learning from an increasing number of noisy teachers. The Hopper and Pick & Place experiments support the same claim in the continuous tasks, with the exception that it can effectively learn from noisy teachers as long as we have at least one good teacher to guide the learning agent.

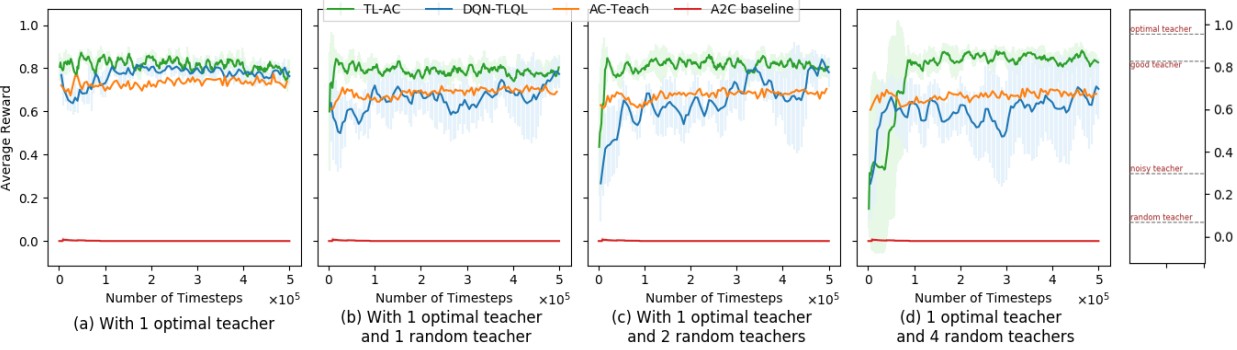

Figure 6: DoorKey results show the robustness of providing 1 optimal teacher and increasing the number of random teachers. AC-Teach (orange lines) and TL-AC (green lines) could always maintain a good performance in all settings, while DQN-TLQL (blue lines) is less steady. Our TL-AC method could quickly converge to an optimal policy, and it always outperforms other benchmark methods.

To further test the robustness of our method, we experiment with a more extreme setting: providing one optimal teacher and increasing the number of random teachers in the teacher set. Figure 6 shows the results of DoorKey domain. TL-AC could outperform the benchmark methods in all 4 settings. In Figure 6(d), the jumpstart of our method could be affected by the increasing number of random teachers, but it adapts quickly and converge to the optimal policy within $1 \times 10^5$ training steps (see Table 1). Meanwhile, the

---

[9]In Hopper, TL-AC could catch up with the A2C baseline with longer training time, around $1 \times 10^6$ training steps, which is not shown in the figure.

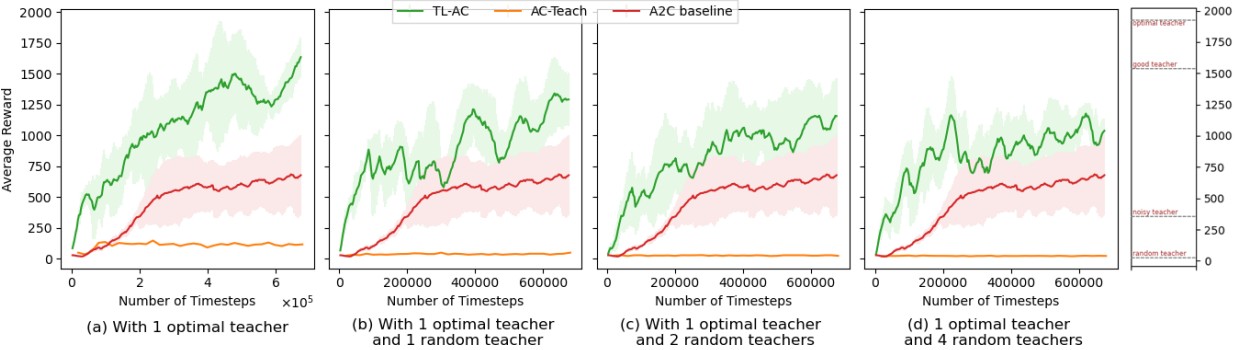

Figure 7: Hopper results show the robustness of providing 1 optimal teacher and increasing the number of random teachers. TL-AC is robust to this setting and outperforms A2C. AC-Teach performs poorly in this task.

DQN-TLQL is unstable with the increasing number of random teachers, and AC-Teach always maintains a steady performance level much lower than TL-AC. Figure 7 shows the results of the Hopper domain. With increasing numbers of random teachers, TL-AC could still benefit from the optimal teacher and surpass the A2C baseline and AC-Teach. Even with an optimal teacher, the overall performance of AC-Teach after $7 \times 10^5$ training steps is still less than 200. The results of Pick & Place can be found in Appendix A.3.3. From this experiment, we conclude that our method is robust to the effect of noisy teachers' advice and could always learn a better policy than the benchmark methods in most cases.

**R2: Can TL-AC incorporate multiple partial teachers with different areas of expertise?**

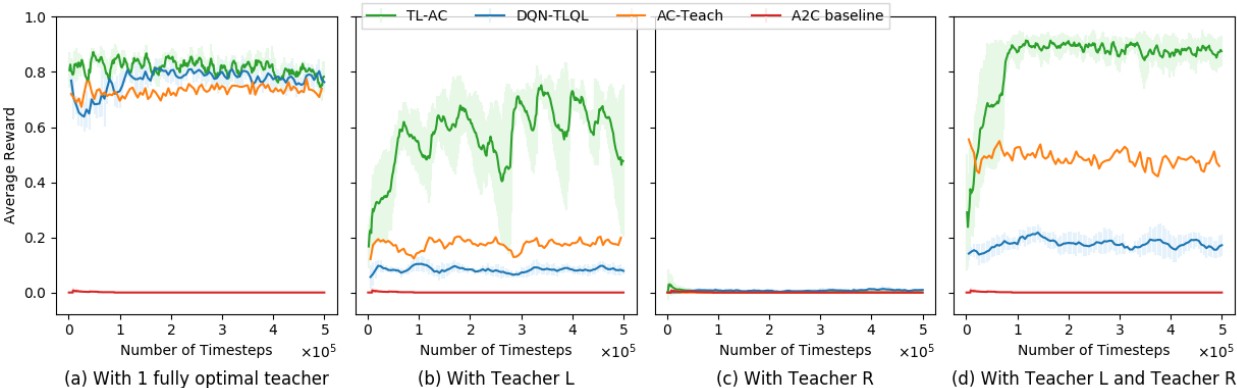

Figure 8: DoorKey results show the capability of incorporating partial teachers. Teacher L or Teacher R could only provide good advice in the left or the right room. With Teacher L, our TL-AC (green lines) method will gradually learn a good policy, while DQN-TLQL (blue lines) and AC-Teach (orange lines) failed to do so. With Teacher R, none of the algorithms could learn and solve the task with the provided teacher. With Teacher L and Teacher R, our TL-AC method could successfully use the advice from those two teachers better than the other methods.

To answer **R2**, we first experiment with the DoorKey task, allowing Teacher L or Teacher R to provide advice to the agent, and results are shown in Figure 8. The target is located in the right room and the door between rooms is locked. To achieve the target, the agent needs to pick up the key, open the door, and then move to the right room. We expected good advice in the left room to be more useful than good advice in the right room. In Figure 8(b), when only providing Teacher L to the agent, TL-AC (green lines) gradually learns a good policy with this partial teacher, while none of the other methods converges to the optimal performance level within the given number of time-steps. When only providing Teacher R, the randomly

suggested actions in the left room will largely disrupt learning. None of the algorithms could eliminate this disturbance and the agent performs poorly in Figure 8(c).

In Figure 8(d), when providing both Teacher L and Teacher R to the agent, TL-AC could learn a near-optimal policy within $1 \times 10^5$ steps, surpassing the performance of using Teacher L only. This suggests that our TL-AC method could use the useful information from Teacher R and converge to a better policy as compared to the settings of providing those teachers separately. While AC-Teach performs better when using both Teacher L and Teacher R, it could not benefit much from Teacher R, as shown in Figure 8(d). Similarly, DQN-TLQL seems to fail to learn with these settings. It could reuse the advice from an optimal teacher trained on the entire task (in Figure 8(a)) but fails to adopt the advice from the teacher of the left room and the teacher of the right room individually (partial but optimal teacher on the sub-tasks). More details and analyses about the reuse frequency of different teachers' policies of our algorithm can be found in Appendix A.1.3.

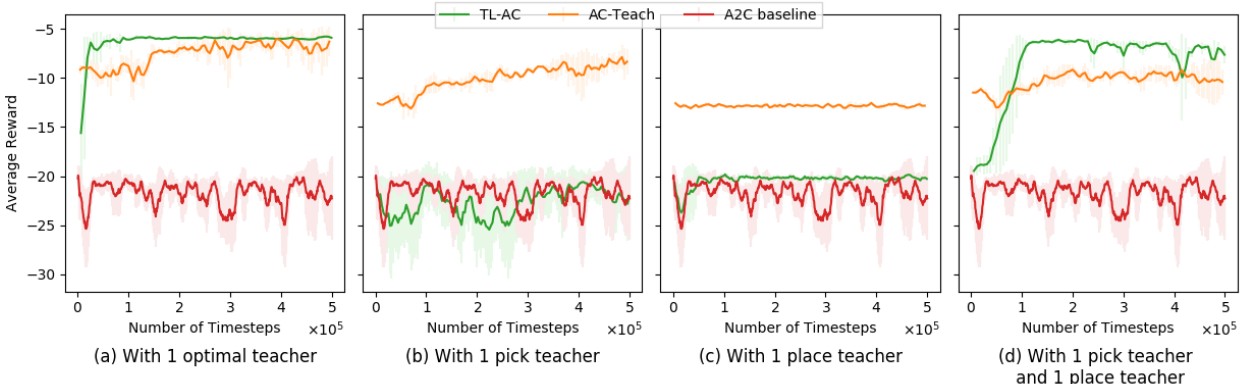

Figure 9: Pick & Place results show the capability of incorporating partial teachers. TL-AC cannot learn a good policy from only a pick or place teacher in (b)(c), but could successfully combine the advice from those two teachers and have a near-optimal performance in (d). AC-Teach(orange lines) has a similar performance level for all scenarios.

We also experiment with the Pick & Place task. Here, the agent first needs to pick up the box, then move it to the target location. We use the same hand-coded *pick teacher* and a *place teacher* to provide action advice as AC-Teach in Kurenkov et al. (2019). The pick teacher only knows how to move toward the box and when to grasp it, and the place teacher only knows how to move the box to the goal. The A2C baseline could not learn a good policy for this task within $1 \times 10^5$ steps training in accordance with prior work (Nair et al., 2018). Even with lower initial performance as compared to AC-Teach, TL-AC could quickly learn a near-optimal policy with a single optimal teacher, or both pick teacher and place teacher, as evident from Figure 9(a) and (d). AC-Teach could learn a better policy from only a pick teacher or a place teacher, while our TL-AC failed to do so, as in Figure 9(b) and (c). With only a pick teacher or a place teacher, the agent still needs to learn a policy from scratch to accomplish the other sub-task (place or pick). Since our low-level policy is based on a simple actor-critic method, which could not learn well in this continuous control task or the sub-task without teacher guidance, therefore our method performs worse than AC-Teach in Figure 9(b) and (c).

AC-Teach could maintain a steady level of performance in all scenarios but could not learn a near-optimal policy with an optimal teacher or a sufficient partial teacher set.

We also analyze the reuse frequency of the pick teacher, place teacher, and agent itself when holding or not holding the box. From Figure 10, we can observe that when holding the box, the frequency of listening to the pick teacher (green line) decreases, and the frequency of listening to the place teacher (orange line) increases. The frequency of reusing those two teachers' policies is about equal between $1 \times 10^5$ to $2 \times 10^5$ training steps, but as training progresses, the agent tends to listen to the Place Teacher's advice, and its performance also converges by this time, as is evident from Figure 9(d). When not holding the box, the frequency of

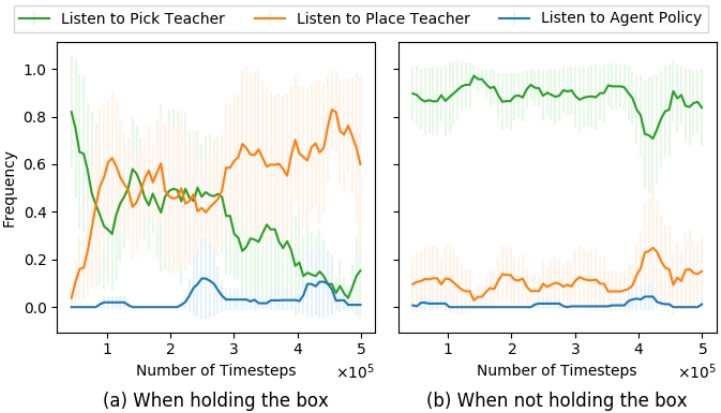

Figure 10: Frequency of listening to the partial teachers and agent in Pick & Place.

listening to the pick teacher is higher than the other two policies during the entire training phase. And the frequency of performing the agent's own policy (blue line) remains relatively low. In summary, TL-AC could efficiently incorporate the advice from multiple teachers with different areas of expertise, as validated by our experiments.

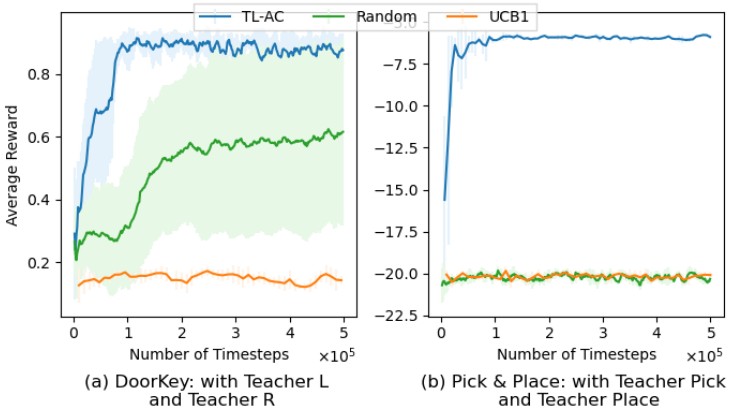

Figure 11: **Using random selection and UCB1 algorithm as high-level policy**

To further verify the efficiency and necessity of our state-based high-level policy structure, we also perform ablation studies with partial teachers in DoorKey and Pick & Place. Figure 11 shows the results of comparing our TL-AC method with (1) Randomly selecting the teachers as the high-level policy denoted as Random; (2) Formalizing the high-level teacher selection as a multi-armed bandit problem, and using the UCB1 algorithm (Auer, 2002) as the high-level policy denoted as UCB1. The teacher is selected at the beginning of each episode based on their estimated return. This structure is similar to the method of Laroche & Feraud (2017). We hypothesize that since UCB1 does not take each state into account, it cannot perform well in both environments. In the DoorKey environment, the random selection has a chance of selecting the proper teacher at each state and reaching a good performance, but with high variance. In Pick & Place, the continuous control task is much more difficult and the random baseline cannot perform well.

## 5 Discussion and Future Work

We introduced a Two-Level Actor-Critic algorithm to address the task of learning from multiple teachers. With the two-level actor and single critic network, this method can effectively incorporate advice from

multiple teachers with different qualities or expertise for both discrete and continuous control tasks for different types of domains.

In discrete tasks like DoorKey, TL-AC could effectively learn from multiple teachers of different qualities, choose the good teachers to listen to, be robust to noise introduced by noisy teachers, and always outperform the baselines. In complex continuous control robotics tasks like Hopper and Pick & Place, TL-AC could still identify good teachers from the pool of teachers. But its capability to recover from bad advice from the sub-optimal teachers depends on the teacher set composition, underlying low-level policy learner quality, and task complexity. For instance, in Pick & Place, when the student agent needs to completely ignore the bad advice from noisy teachers and develop its own policy, or when needs to accomplish the "subtask" of moving the picked box to the target location without teachers' advice, TL-AC is not able to outperform AC-Teach. For all the different domains considered in our work, as long as there is full coverage of good advice, TL-AC effectively learns to listen to it and filter out the noisy guidance from other teachers.

To summarize, advantages of using TL-AC include the following:

1. being lightweight and having a simple structure,
2. easily switching between policies at every time-step to incorporate the best teacher's advice,
3. being adaptable to a wide variety of tasks,
4. simple incorporation into any actor-critic algorithm, and
5. working with both full and partial teachers.

One limitation of TL-AC includes requiring the teacher to always be available. Second, its performance can be poor when only noisy teachers are available.

As future work, first, conducting experiments with human teachers remains an interesting direction. Second, investigating the possibility of incorporating uncertainty and confidence-related schemes into this framework remains another open direction. Third, relaxing the assumption that teachers are always available and introducing a budgeted version of TL-AC where teachers can provide advice a limited number of times are immediate future directions that we aim to pursue.

### Acknowledgments

Part of this work has taken place in the Intelligent Robot Learning (IRL) Lab at the University of Alberta, which is supported in part by research grants from the Alberta Machine Intelligence Institute (Amii); a Canada CIFAR AI Chair, Amii; Compute Canada; Huawei; Mitacs; and NSERC.

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

# A   Additional Experiments Details

## A.1   DoorKey Experiment

### A.1.1   DoorKey: Teacher Set

The MiniGrid-DoorKey-7x7-v0 task has sparse reward, the agent could only get reward after successfully reaching the target square. The success requires the agent to perform sequences of behaviors: pick up the key, open the door, and reach the target square. This task is a good testbed to evaluate whether our method could incorporate multiple teachers with different expertise. Since the task difficult to learn for the A2C baseline, we obtain a optimal policy with PPO algorithm, which trained for $1 \times 10^6$ steps. The training reward of this agent is 0.96, testing mean reward of 1000 episode is 0.77 and standard deviation is 0.38.

With this optimal policy, we construct teacher set as follows:

1. Optimal Teacher: Always give advice according to the optimal policy.
2. Good Teacher: Give advice according to the optimal policy with a probability of 0.8, and give advice randomly with a probability of 0.2. The average performance is 0.83.
3. Noisy Teacher: Give advice according to the optimal policy with a probability of 0.2, and give advice randomly with a probability of 0.8. The average performance is 0.30.
4. Teacher L: Give advice according to the optimal policy when at the left room, and give random advice when at the right room. It knows how to pickup the key and open the door.
5. Teacher R: Give advice according to the optimal policy when at the left room, and give random advice when at the right room. It knows how to navigate to the target square.
6. Random Teacher: Always give advice randomly. The average performance is 0.007.

### A.1.2   DoorKey: Jumpstart

Table 1 shows the jumpstarts of benchmark methods in DoorKey.

Table 1: Jumpstart of benchmark methods with providing different teacher set

| Teacher Set | TL-AC | AC-Teach | DQN-TLQL |
|---|---|---|---|
| 3 Good Teachers | 0.85 | 0.62 | 0.83 |
| 2 Good Teachers + 1 Noisy Teacher | 0.80 | 0.66 | 0.76 |
| 1 Good Teachers + 2 Noisy Teachers | 0.79 | 0.69 | 0.67 |
| 3 Noisy Teachers | 0.13 | 0.63 | 0.26 |
| 1 Optimal Teacher | 0.81 | 0.72 | 0.77 |
| 1 Optimal Teacher + 1 Random Teacher | 0.73 | 0.58 | 0.64 |
| 1 Optimal Teacher + 2 Random Teachers | 0.52 | 0.60 | 0.26 |
| 1 Optimal Teacher + 4 Random Teachers | 0.31 | 0.59 | 0.26 |
| Full Optimal Teacher | 0.81 | 0.72 | 0.77 |
| Teacher L | 0.23 | 0.74 | 0.05 |
| Teacher R | 0.03 | 0.0 | 0.002 |
| Teacher L + Teache R | 0.29 | 0.57 | 0.0014 |

Note: In situations where our TL-AC has a lower jumpstart, it could quickly catch up in the later training stages and surpass the performance of DQN-TLQL and AC-Teach.

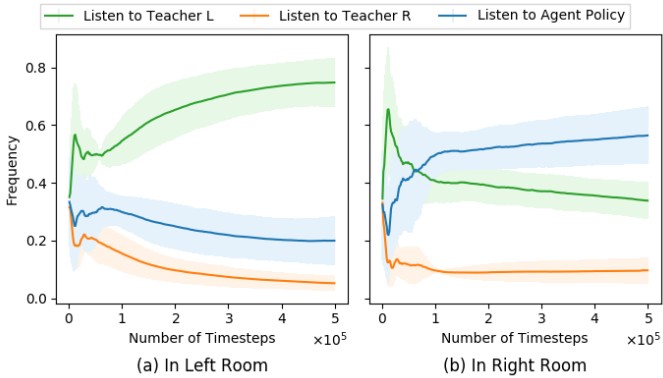

Figure 12: **Frequency of listening to the partial teachers and agent in DoorKey.**

### A.1.3   DoorKey: Reuse Frequency of Teacher L and Teacher R

We also check the reuse frequency of the Teacher L, Teacher R, and the agent itself in the left room and right room. From Figure 12, we could observe that at the beginning, the frequency of listening to Teacher L (green line) or performing agent's own policy (blue line) are about equal (due to initial exploration). In the left room, the frequency of reusing Teacher L's policy first increases to reuse the useful information, then decreases as training progresses, and frequency of performing agent's own policy first increases and remain at a steady frequency level as the agent learns over time. We should focus our attention to the region before $1 \times 10^5$ steps for this behavior because (as is evident from Figure 8) the TL-AC agent converges to the optimal policy by this step. The sub-task of the right room is to go to the target square, which is simpler task for agent to learn. In the right room, the frequency of Teacher L's policy drops as training goes, and the agent gradually prefer to perform its own policy (blue line) rather than listen to Teacher L or Teacher R.

### A.1.4   DoorKey: Performance at the Beginning Stage

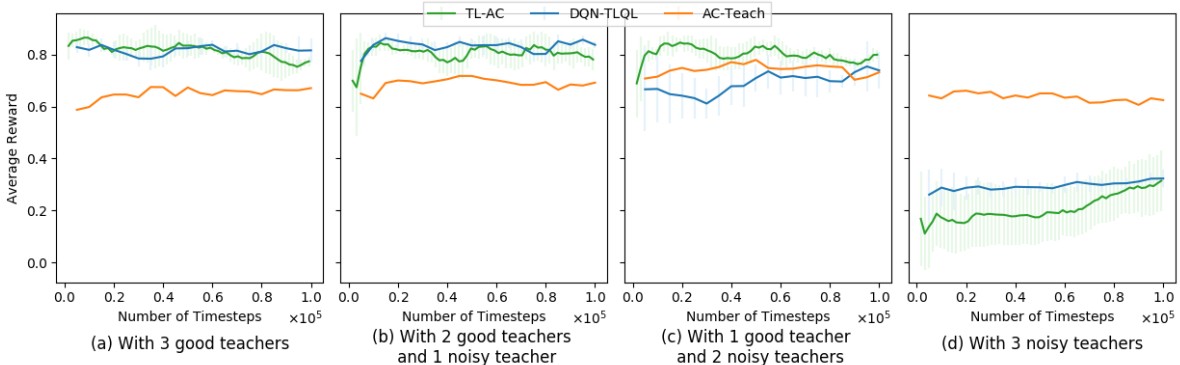

Figure 13: **First $10^5$ steps of the results show in Figure 3**

### A.2   Hopper Experiment

### A.2.1   Hopper Teacher Set:

This task is a good testbed to evaluate whether our method could handle high-dimensional continuous state and action space. Since the DQN-TLQL could not be applied to this continuous action task, we only compare with the A2C baseline and the AC-Teach method in the following experiments.

For the teacher set construction, we first obtain an optimal policy with SAC algorithm, which trained for 1,000,000 steps. The mean reward of this policy over 1000 testing episode is 1928 and standard deviation is 646.

With this optimal policy, we construct teacher set as follows:

1. Optimal Teacher: Always give advice according to the optimal policy.
2. Good Teacher: Give advice according to the optimal policy with a probability of 0.8, and give advice randomly with a probability of 0.2.
3. Noisy Teacher: Give advice according to the optimal policy with a probability of 0.2, and give advice randomly with a probability of 0.8.
4. Random Teacher: Always give advice randomly.

### A.2.2 Hopper: Jumpstart

Table 2 shows the jumpstarts of benchmark methods in Hopper. Our TL-AC method could always have a higher jumpstart compared to AC-Teach.

Table 2: Jumpstart of benchmark methods with providing different teacher set

| Teacher Set | TL-AC | AC-Teach |
|---|---|---|
| 3 Good Teachers | 60.29 | -14.95 |
| 2 Good Teachers + 1 Noisy Teacher | 7.82 | -26.13 |
| 1 Good Teachers + 2 Noisy Teachers | 0.9 | -33.17 |
| 3 Noisy Teachers | -25.09 | -29.98 |
| 1 Optimal Teacher | 36.61 | -23.07 |
| 1 Optimal Teacher + 1 Random Teacher | 19.34 | -22.59 |
| 1 Optimal Teacher + 2 Random Teachers | -7.71 | -23.51 |
| 1 Optimal Teacher + 4 Random Teachers | -11.84 | -30.02 |

### A.3 Pick & Place Experiment

### A.3.1 Pick & Place Teacher Set:

This task is for evaluating whether our method could handle high-dimensional continuous control task. Since the DQN-TLQL could not be applied to this continuous action task, we only compare with the A2C baseline and the AC-Teach method in the following experiments.

For the teacher set construction, we use the same optimal policy as Kurenkov et al.'s work Kurenkov et al. (2019): first directly move to the box and grasp it, then move towards the target location and drop it. The mean reward of this policy over 1000 testing episodes is -5.83 and standard deviation is 1.23.

With this optimal policy, we construct teacher set as follows:

1. Optimal Teacher: Always give advice according to the optimal policy.
2. Good Teacher: Give advice according to the optimal policy, but with addition action noise of 0.01. The mean reward of this teacher is -5.96 and standard deviation is 1.26 over 1000 testing episodes.
3. Noisy Teacher: Give advice according to the optimal policy, but with addition action noise of 0.1. The mean reward of this teacher is -10.82 and standard deviation is 5.36 over 1000 testing episodes.
4. Pick Teacher: Give advice according to the optimal policy when not hold the box, will move towards the box and pick it up.

5. Place Teacher: Give advice according to the optimal policy when hold the box, will move towards the target location.
6. Random Teacher: Always give advice randomly. The mean reward of this teacher is -20.62 and standard deviation is 4.31 over 1000 testing episodes.

### A.3.2 Pick & Place: Jumpstart

Table 3 shows the jumpstarts of benchmark methods in Pick & Place. When having a teacher set of multiple good teachers, our TL-AC method could have a higher jumpstart compared to AC-Teach. When having more noisy or random teachers in the teacher set, the AC-Teach method will have a higher jumpstart.

Table 3: Jumpstart of benchmark methods with providing different teacher set

| Teacher Set | TL-AC | AC-Teach |
|---|---|---|
| 3 Good Teachers | 13.59 | 10.60 |
| 2 Good Teachers + 1 Noisy Teacher | 12.80 | 11.73 |
| 1 Good Teachers + 2 Noisy Teachers | 10.53 | 10.94 |
| 3 Noisy Teachers | 7.79 | 10.72 |
| 1 Optimal Teacher | 6.59 | 10.49 |
| 1 Optimal Teacher + 1 Random Teacher | -1.15 | 9.56 |
| 1 Optimal Teacher + 2 Random Teachers | 4.15 | 9.17 |
| 1 Optimal Teacher + 4 Random Teachers | 1.75 | 9.10 |
| Full Optimal Teacher | 6.59 | 10.49 |
| Teacher Pick | 0.21 | 9.62 |
| Teacher Place | 0.36 | 9.59 |
| Teacher Pick + Teach Place | 2.72 | 9.58 |

### A.3.3 Pick & Place: Robustness

While in the Pick & Place task, the performance of our method could be affected with the increasing number of random teachers as in Figure 14. From (a) to (d), the mean reward drops and the standard deviation also fluctuated. Even with such fluctuation, TL-AC could still outperform AC-Teach in all scenarios.

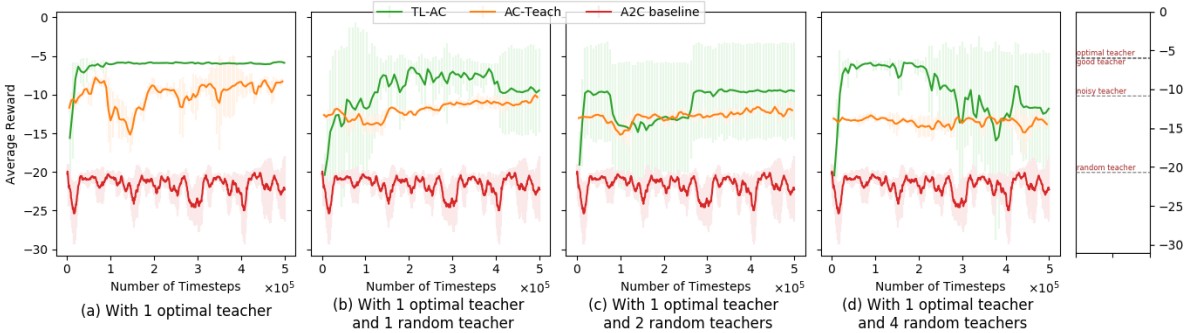

Figure 14: Pick and Place results show robustness of providing 1 optimal teacher and increasing the number random teachers. AC-Teach (orange lines) is robust to the increasing number of random teachers. Our TL-AC (green lines) could be affected with increasing number of random teachers, but still performs better than AC-Teach.

## B Hyperparameter Settings

### B.1 Hyperparameters for DoorKey Experiment

Table 4: Hyperparameters of A2C in DoorKey

| | |
|---|---|
| Number of envs | 4 |
| Numer of steps per update | 5 |
| Buffer size | 20 |
| Number of steps | 5e5 |
| Policy | 'MlpPolicy' |
| Entropy coefficient | 0.0 |
| Optimizer | RMSpropTFLike |
| RMSProp epsilon | 1e-5 |
| Value function coefficient | 0.5 |
| Maximum value for gradient clipping | 0.5 |
| Gamma | 0.99 |
| GAE coefficient | 0.9 |
| Normalize advantage | True |
| Learning rate | 7e-4 |
| Hidden Layer | pi=[64, 64], vf=[64, 64] |

Table 5: Hyperparameters of TL-AC in DoorKey

| | |
|---|---|
| Number of envs | 1 |
| Numer of steps per update | 16 |
| Buffer size | 16 |
| Number of steps | 5e5 |
| Epochs | 5 |
| Target KL | 0.02 |
| Clipping parameter | 0.2 |
| Policy | 'MlpPolicy' |
| Entropy coefficient | 0.0 |
| Optimizer | RMSpropTFLike |
| RMSProp epsilon | 1e-5 |
| Value function coefficient | 0.5 |
| Maximum value for gradient clipping | 0.5 |
| Gamma | 0.99 |
| GAE coefficient | 0.85 |
| Normalize advantage | True |
| Learning rate | 7e-4 |
| Hidden Layer | high-level: pi=[64, 64], vf=[64, 64] low-level: pi=[64, 64], vf=[64, 64] |

Table 6: Hyperparameters of DQN-TLQL in DoorKey

| | |
|---|---|
| Number of envs | 1 |
| Numer of steps per update | 100 |
| Tau | 0.2 |
| Replay memory size | 10,000 |
| Target update interval | 100 |
| Exploration rate | 1→0.01 |
| Exploration fraction | 0.1 |
| Number of steps | 5e5 |
| Policy | 'MlpPolicy' |
| Maximum value for gradient clipping | 10 |
| Gamma | 0.99 |
| Learning rate | 0.0001 |
| Optimizer | Adam |
| Hidden Layer | high-level: pi=[64, 64], vf=[64, 64] low-level: pi=[64, 64], vf=[64, 64] |

## B.2   Hyperparameters for Hopper Experiment

Table 7: Hyperparameters of A2C in Hopper

| | |
|---|---|
| Number of envs | 4 |
| Numer of steps per update | 8 |
| Buffer size | 32 |
| Number of steps | 1e6 |
| Policy | 'MlpPolicy' |
| Log std init | -2 |
| Ortho init | False |
| Entropy coefficient | 0.0 |
| Optimizer | RMSpropTFLike |
| RMSProp epsilon | 1e-5 |
| Use gSDE | True |
| gSDE sample frequency | 4 |
| Value function coefficient | 0.4 |
| Maximum value for gradient clipping | 0.5 |
| Gamma | 0.99 |
| GAE coefficient | 0.9 |
| Normalize advantage | False |
| Learning rate | linear_schedule (0.00096, 5e-4) |
| Hidden Layer | pi=[64, 64], vf=[64, 64] |

## B.3   Hyperparameters for Pick & Place Experiment

Table 8: Hyperparameters of TL-AC in Hopper

| Number of envs | 4 |
|---|---|
| Numer of steps per update | 8 |
| Buffer size | 32 |
| Number of steps | 1e6 |
| Epochs | 10 |
| Target KL | linear_schedule (0.04, 0.01) |
| Clipping parameter | linear_schedule (0.3, 0.1) |
| Policy | 'MlpPolicy' |
| Log std init | low:-2, high: 0.0 |
| Ortho init | low: False, high: True |
| Entropy coefficient | 0.0001 |
| Optimizer | RMSpropTFLike |
| RMSProp epsilon | 1e-5 |
| Use gSDE | True |
| gSDE sample frequency | 4 |
| Value function coefficient | 0.45 |
| Maximum value for gradient clipping | 0.5 |
| Gamma | 0.99 |
| GAE coefficient | 0.9 |
| Normalize advantage | True |
| Learning rate | linear_schedule (5e-4, 3e-5) |
| Hidden Layer | high-level: pi=[64, 64], vf=[64, 64] low-level: pi=[64, 64], vf=[64, 64] |

Table 9: Hyperparameters of A2C in Pick & Place

| Number of envs | 4 |
|---|---|
| Numer of steps per update | 5 |
| Buffer size | 20 |
| Number of steps | 5e5 |
| Policy | 'MlpPolicy' |
| Entropy coefficient | 0.0 |
| Optimizer | RMSpropTFLike |
| RMSProp epsilon | 1e-5 |
| Value function coefficient | 0.5 |
| Maximum value for gradient clipping | 0.5 |
| Gamma | 0.99 |
| GAE coefficient | 0.9 |
| Normalize advantage | True |
| Learning rate | 7e-4 |
| Hidden Layer | pi=[64, 64], vf=[64, 64] |

Table 10: Hyperparameters of TL-AC in Pick & Place

| | |
|---|---|
| Number of envs | 4 |
| Numer of steps per update | 16 |
| Buffer size | 96 |
| Number of steps | 5e5 |
| Epochs | 20 |
| Target KL | linear_schedule(0.03, 0.02) |
| Clipping parameter | 0.2 |
| Policy | 'MlpPolicy' |
| Log std init | low:-2, high: 0.0 |
| Ortho init | low: False, high: True |
| Entropy coefficient | 0.0001 |
| Optimizer | RMSpropTFLike |
| RMSProp epsilon | 1e-5 |
| Use gSDE | True |
| gSDE sample frequency | 4 |
| Value function coefficient | 0.45 |
| Maximum value for gradient clipping | 0.5 |
| Gamma | 0.99 |
| GAE coefficient | 0.9 |
| Normalize advantage | True |
| Learning rate | 0.00015 |
| Hidden Layer | high-level: pi=[64, 64], vf=[64, 64] 
 low-level: pi=[64, 64], vf=[64, 64] |

