# OpenReview forum: "Two-Level Actor-Critic Using Multiple Teachers"
_TMLR — Accepted by TMLR_

### Review · Reviewer_bExB · 2023-05-30

**Summary Of Contributions:**

The authors present a simple technique to learning an agent under advice from multiple teachers. This two-level technique essentially choses either a teacher’s action to execute or a learned policy’s action to execute. Rewards then propagate through off-policy RL to ensure that the learned agents learns from the teachers’ actions. The authors demonstrate results on MiniGrid, locomtion, and simple pick and place tasks along with analysis of the effects of noisy teachers.

**Audience:**

Yes

**Claims And Evidence:**

Yes

**Requested Changes:**

The main drawback of this paper is its lack of methodological contribution. While the authors present some new experiments involving noisy teachers, these experiments are not comprehensive enough to merit acceptance of this paper for its analysis and experiments.

I would recommend framing this paper less as a “new method,” rather an analysis of existing methods (TLQL, AC-Teach, etc.). Then the authors can also experiment with swapping out the underlying algorithms for more modern SOTA algorithms like SAC/PPO/etc. and investigating the effects of these changes on performance on existing tasks (among other design choices), and then the resulting algorithm can be the accumulation of the insights discovered through this analysis. This format would be more valuable for the RL community.

 If the authors choose to follow this analysis paper route, I would also recommend experimenting with more teachers and/or with more difficult environments and control tasks.

**Strengths And Weaknesses:**

********************Strengths:********************

- ****************Clarity:**************** The paper is very clear and easy to read.
- ****************Results:**************** Results are somewhat promising and demonstrate robustness to noisy teachers.

**********************Weaknesses:**********************

- ****************Writing:**************** The authors should give an intuition of their approach before then explaining it in detail in Section 4.2. Right now there’s no intuition for why they have set up the two-level hierarchy in the exact manner, perhaps an overview paragraph at the top of 4.2 before then talking about the algorithm would be the best place to put it.
- **Contribution**: Unfortunately I don’t think this paper presents a significant enough contribution for this conference. They essentially use Li 2019b (Two level q-learning) but modify it to use A2C as the base algorithm so that it can work for continuous control. While they add analysis regarding noisy teachers and such, the paper is mainly framed as a method paper and I don’t think that swapping out the base algorithm of an existing 2-level setup is a significant methodological contribution.
- **Baselines**: A missing, simple baseline is one that does flat RL with advice from teachers incorporated. That would be, for example, one in which the policy loss is modified by adding a distribution loss term (e.g. KLD loss) that constrains the learned policy to be somewhat close to all of the teacher’s output actions.
- **************Analysis:************** Why is this TL-AC algorithm better than AC-Teach and DQN-TLQL in certain instances? The analysis could be improved in the experiments to make this more clear. The authors, for example, say “we conclude that our method is robust to the effect of noisy teachers’ advice,” but don’t say why it’s more robust.

**************Minor Issues:**************

- L20 in Alg1: “transcation” → “transaction”

---

### Review · Reviewer_n2pd · 2023-06-06

**Summary Of Contributions:**

This paper proposes an approach to reinforcement learning (RL) in the presence of multiple teachers, where a teacher is understood to be a policy. Thus, the RL problem of interest consists of an agent interacting with a Markov decision process (MDP), where the agent also has access to a collection of $N$ policies, $\{\pi_{e_1}, \ldots, \pi_{e_N}\}$, describing the teachers behaviors. The key premise of the work is that the teachers may have different sets of expertise: one teacher might be effective in one region of the state space, while ineffective elsewhere (in the given example, one teacher may be good at mopping, while another may be best at cleaning counter tops). Thus, the key question facing the RL agent is whether---and when---to trust each teacher, rather than default to their own behavior. The paper proposes Two-Level Actor-Critic (TL-AC, Algorithm 1), an algorithm that makes use of teacher action advice to learn efficiently. Experiments center around two research questions: (R1) how does TL-AC deal with multiple _sub-optimal_ teachers, and (R2) Can TL-AC make use of teachers with differing expertise? The empirical study is broad and deep, covering a variety domains including a discrete control task (DoorKey), as well as continuous control tasks (Hopper). Baselines are A2C, DQN with Two-level Q-learning, and AC-Teach. The first experiments explore the impact of the presence of more noisy teachers: for instance, Figure 3 and 4 show how performance changes as teachers are changed between three good teachers (subfigures 3a and 4a), and three noisy teachers (subfigures 3d and 4d). The second set of experiments address R2, and explore whether the TL-AC can learn effectively even as teachers have diverse expertise.

**Audience:**

Yes

**Claims And Evidence:**

Yes

**Requested Changes:**

Below are some suggested writing changes and other low-level questions:
- Typo in Algorithm 1: Line 20 "transcation" --> "transaction". Also, I would call this an "experience" rather than a transaction.
- I recognize previous literature has referred to these policies as "teachers", but I find this slightly confusing. The policies are _any_ policy, as the paper states. For this reason I find some conceptual overlap with work on transfer, as in the work by Barreto et al. discussed above.
- Why are the confidence intervals for A2C so thin, and only appear around specific points in the results? Was the sampling rate different?
- I found that some of the notation in Algorithm 1 was confusing. At line 15, what is the argmax over? What is being maximized? In lines 7-10, the advantage changes from $s_t$ to $s$. In line 6, you label __M__, but don't use this notation again.
- It looks as though both $R$ and $r$ are used as the reward function ($R$ appears in the background section, but $r(s,a)$ in Section 4, for example in Equation 1 or 2).
- The background section is so short I don't believe it needs to be its own section. Consider folding both related work and background into the introduction as subsections?
- Also, I would prefer the related work section of this paper come _after_ the main methods section (currently Section 4 Two-Level Actor-Critic Using Multiple Teachers). This will allow you to contrast details of your approach.

**Strengths And Weaknesses:**

[Strengths]

This paper possesses many strengths. The contributions of this paper are quite clear: a new algorithm for a specific setting of RL in which a set of policies are available. This is a general enough setting that I believe the algorithm and experiments will be of interest to the community. Determining usefulness of this algorithm is framed by research questions R1 and R2, which are examined in great detail in the experiments. Experiments are well designed to isolate interesting trends, such as how performance degrades across learning algorithms as the number of noisy expert policies increases. Domains are well chosen, as are baselines. Assumptions are stated plainly.

[Weaknesses]

The biggest weakness of the work is in its restricted scope. The learning setting studied is stylized, and as such, the proposed algorithm is most useful under specific assumptions (the presence of many reference policies that can be queried freely). I don't find this weakness to be substantial enough to prevent publication. The only other minor weakness is that there is a body of literature that focuses on the RL problem in the presence of reference policies, such as "Successor features for transfer in reinforcement learning" by Barreto et al., or "Policyblocks" by Pickett and Barto. Both works (and the neighborhood of work surrounding them) consider how to best learn given a set of policies. These policies might be sub-optimal. The most crucial idea I felt was missing from the discussion of the present paper was "Generalized Policy Improvement" (or GPI) from the Barreto et al. work---this proposes a way to improve behavior over a set of policies. In follow up work (Barreto et al. 2018), they extend these ideas to Deep RL. I wonder how the authors find this kind of approach fairs in comparison to TL-AC?


References:

Barreto, André, et al. "Successor features for transfer in reinforcement learning." Advances in neural information processing systems 30 (2017).

Barreto, Andre, et al. "Transfer in deep reinforcement learning using successor features and generalised policy improvement." International Conference on Machine Learning. PMLR, 2018.

Pickett, Marc, and Andrew G. Barto. "Policyblocks: An algorithm for creating useful macro-actions in reinforcement learning." ICML. Vol. 19. 2002.

---

### Review · Reviewer_HEyH · 2023-06-08

**Summary Of Contributions:**

The authors tackle an online RL setting where several advising policies called teachers are provided. These teachers do not need to be expert, even though their added value depend on it. In order to take advantage of the teachers, the algorithm proposes a bi-level decision making where it first makes the decision of which agent (teacher or low-level policy) to select as control, and second, if the low-level policy has been selected, which action to take. While pretty straightforward, this approach is sound and novel as far as I know. The authors demonstrate the advantage of their approach as compared to a few baselines taking advantage or not of the teachers, in a few settings (mix of good/bad teachers), and in a few environments.

**Audience:**

Yes

**Broader Impact Concerns:**

no concern

**Claims And Evidence:**

Yes

**Requested Changes:**

I expect the authors to improve the submission in order to fix the points mentioned above.
I do not think that there is anything unfixable, but these would require a significant revision of the submission.

Also, I would be grateful if they could add Reinforcement Learning Algorithm Selection [Laroche2019] in their competitors. The principle is pretty simple too: at the start of every trajectory, a high-level policy (which instantiated as a multi-armed bandit) selects a policy to be in control of the full trajectory (this is therefore a trajectory agent selection and not a timestep agent selection as in the submission). The algorithm portfolio would therefore be the set of teachers plus a learning algorithm (A2C) that learns to solve the task. I think that the addition of [Laroche2018] in the competitors would highlight the need for a state-based expert selection.

[Laroche2018] Laroche, Romain, and Raphael Feraud. "Reinforcement Learning Algorithm Selection." International Conference on Learning Representations 2018.

**Strengths And Weaknesses:**

Strengths:
- simple, sound, and novel algorithmic solution
- the empirical study covers well the various set-ups I could imagine

Weaknesses:
- the last paragraph of the first page that intends to motivate the setting is not convincing at all. It's unrealistic and not a setting where we would have naturally human experts and even less where there is a reason for needing several complementary experts. I'm sure there could be a better motivating example. (if not, I would suggest to just remove it)
- the formalization is approximative (details follow).
- the positioning is fuzzy. I do not know well the competitors and I did not have a clue of how they work by reading the submission. For instance, it seems that the novelty as compared to Li2019b is only the extension to continuous action spaces, but I am not sure. Is it so?
- some empirical results are poorly analyzed (details follow).

Approximative formalization:
- equation (4): what is $r$? The reward function is called $R$. I don't understand what this equation is supposed to show/mean. Or maybe do you mean that there exists an MDP where the action space is $E$ instead of $A$ and its reward function is defined as follows?
- equation (5): you need to show the dependencies here. It does not seem right. If the values of the high and low level policies are equal, what is the use of the teachers?
- algorithm 1 line 4: why is $m$ passed to collect function? Isn't $\pi_{low}$ part of $E$, why is it passed to collect?
- algorithm 1 line 23: what does it mean that $B$ is full? It means that at the next iteration of line 3 loop, $B$ will still be full?

Empirical analysis:
- Figure 3: this is weird how none of the algorithm seems to learn anything (most of the curves are constant over time). You need to show how the algorithms reached their respective performances.
- Figure 3: What are the teachers' performance? It should be visible on the plot.
- Figure 4: It shows that TL-AC high policy tends to learn to never use $\pi_{low}$ because it's low-performing at start. You would need a reset (like they do in [Laroche2018].
- Figure 4: shouldn't AC-teach work okay as long as there is a good teacher to select? Again, it would help the understanding to see the performance of the teachers.

Minor comments and typos:
- $log$ => $\log$
- value network squared loss should start with $L(\theta')$
- algorithm 1 is displayed 2 pages too late.

---

### Decision · Action_Editors · 2023-08-19

**Recommendation:** Accept with minor revision

**Comment:**

The paper studies an online RL setting where an agent has to select an action under the advice of multiple teachers.
The proposed approach is simple, sound, and original.
The empirical analysis is extensive and well-executed.
However, the paper has some weaknesses (approximative formalization, missing references to related works, better highlighting the contribution, and improving the analysis of the experimental results) that the authors have partially addressed in their rebuttals.
The authors now have to implement these changes in the new revision of their paper.

**Audience:**

The topic of this paper is of interest to a significant part of the TMLR's audience.

**Claims And Evidence:**

The claims made in the paper are supported by accurate empirical analysis.